# Study on Microstructure and Fatigue Properties of FGH96 Nickel-Based Superalloy

**DOI:** 10.3390/ma14216298

**Published:** 2021-10-22

**Authors:** Yishan Bai, Shanglei Yang, Minqi Zhu, Cong Fan

**Affiliations:** 1School of Materials Engineering, Shanghai University of Engineering Science, Shanghai 201620, China; baiyishan1997@163.com (Y.B.); zhuminqi1997527@163.com (M.Z.); fc19960802@163.com (C.F.); 2Shanghai Collaborative Innovation Center for Laser Advanced Manufacturing Technology, Shanghai University of Engineering Science, Shanghai 201620, China

**Keywords:** synchrotron radiation X-ray imaging, fatigue performance, fatigue crack, FGH96 alloy, microstructure

## Abstract

In this study, using synchrotron radiation X-ray imaging, the microstructure, tensile properties, and fatigue properties of FGH96 nickel-based superalloy were tested, and the fatigue damage mechanism was analyzed. An analysis of the experimental results shows that the alloy structure is dense without voids or other defects. It was observed that the primary γ′ phase is distributed on the grain boundary in a chain shape, and the secondary γ′ phase is found inside the crystal grains. The X-ray diffraction (XRD) pattern indicates that no other phases were seen except for the γ and γ′ phases. The tensile strength of the alloy is 1570 MPa and the elongation is 12.1%. Using data fitting and calculation, it was found that the fatigue strength of the alloy under the condition of 5 × 10^6^ cycles is 620.33 MPa. A fatigue fracture has the characteristics of secondary crack, cleavage step, fatigue stripe, tire indentation, and dimple. The fracture is a mix of cleavage fracture and ductile fracture. Through a three-dimensional reconstruction of the alloy synchrotron radiation imaging area, it was found that the internal defects are small and mostly distributed at the edge of the sample. The dimple morphology is formed by cavity aggregation and cavity germination resulting from defects in the material itself, fracture of the second-phase particles, and separation of the second-phase particles from the matrix interface. By analyzing the damage mechanism of fatigue fractures, it is concluded that the cleavage step is formed by the intersection of cleavage planes formed by branch cracks, with the main crack of the confluence extending forward to form a cleavage fracture. The crack propagation path was also analyzed, and under the action of cyclic load and tip passivation, the crack shows Z-shaped propagation.

## 1. Introduction

The turbine disk is one of the most important components of the aero engine, and its working environment requires a high level of performance [1,2]. Powder metallurgy FGH96 nickel-based superalloy is used in the manufacture of turbine disks in the Chinese aviation industry due to its uniform structure, fine grain strengthening, high tensile strength, and high fatigue properties [3,4,5,6]. In the practical application of turbine disks, the fluctuation of air flow and the sharp changes in pressure produced by aircraft take-off and landing means there are higher requirements for the fatigue strength of turbine disks, and it is necessary to ensure that the turbine disk has sufficient anti-fatigue ability [7,8,9]. There is still a large gap between China and other countries in the manufacturing, microstructure and properties, service, and fatigue reliability of aviation turbine disk powder nickel-based superalloy, particularly with regard to the manufacturing process, impurity control, and fatigue properties [10,11,12].

Scholars at home and abroad have done much research on the fatigue properties of nickel-based superalloys. Hu et al. [13] studied the influence of the location of inclusions on the low-cycle fatigue life of the FGH96 turbine disk. The experimental results show that surface defects such as surface scratches and surface inclusions have a greater influence on the fatigue life, while the fatigue life of inclusions in the sample is usually longer. Jiménez et al. [14] studied the crack initiation and propagation of IN718 nickel-based superalloy under low-cycle fatigue by means of auxiliary in situ diffraction and phase contrast techniques. The results show that the propagation of short cracks in the microstructure is determined by the slip transfer path through the pre-existing grain boundary.

The synchrotron radiation X-ray imaging technology of the Shanghai light source can obtain the three-dimensional structure of defects without destroying the material structure [15,16]. The large advantage in the internal detection of materials has not been widely studied in the field of engineering application and applied basic research. There are few international and domestic researchers, and the test materials that have been studied are mostly aluminum alloy and titanium alloy [17,18,19].

In this paper, the size, shape, and distribution of the defects in FGH96 nickel-based superalloy are analyzed based on synchrotron radiation X-ray imaging. Testing the fatigue performance of the PM (powder metallurgy) FGH96 nickel-based superalloy turbine disk, determining the fatigue life limit of the material, and studying the fatigue damage behavior of the alloy have important basic theoretical significance for the development of new PM superalloy turbine disk materials and the safe operation of civil aircraft.

## 2. Experimental Materials and Methods

The experimental material is the FGH96 nickel-based superalloy aviation turbine disk, which was provided by Aviation Industry Corp (Beijing, China). The element content of the material is shown in Table 1.

The fatigue test dimensions were designed according to the national standard GB/t3075-2008, as shown in Figure 1a, the tensile test dimensions were designed according to GB/T228-2002, as shown in Figure 1b, and the synchrotron radiation X-ray computed tomography (SR-μCT) dimensions were designed according to the requirements of the Shanghai light source test samples, as shown in Figure 1c. The three samples were obtained by wire cutting. The microstructure observation, hardness test, fatigue test, fatigue, and tensile specimen fracture observation test were carried out on the material. The test methods and test equipment are shown in Table 2 below.

The internal defects of the material were detected at BL13W1 of Shanghai synchrotron radiation source (see Figure 2). The SR-μCT sample was placed 30 cm away from the 4.0 SCMOS detector of a Hamamatsu flash lamp. To ensure penetration, the photon energy was 42 keV. The space pixel size was 0.65 μm. The exposure time was 4 s, and the imaging area is shown in Figure 1d. In the process of X-ray imaging, the sample is usually rotated 180° around the vertical axis, the rotation angle increment is 0.25°, and 720 projections can be collected at one time. Pitre and P3B software (PITRE V. 3.0, 2013, Shanghai Institute of Applied Physics, Chinese Academy of Sciences) were used to convert the data into 8-bit slices, and Avizo software (Avizo 9.5, 2019, Thermo Fisher Scientific, Waltham, MA, USA) was used to reconstruct the internal defects.

## 3. Result Analysis and Discussion

### 3.1. Microstructure Analysis

Figure 3 shows the microstructure of FGH96 nickel-based superalloy. Figure 3a shows the microstructure of 500 folds of the super depth of field. It can be seen from the figure that the grains are evenly distributed, the structure is dense, and there are no voids or cavities. According to the data, the primary γ′ phase diameter is approximately 200 nm, the secondary γ′ phase diameter is 50–200 nm, and the three times γ′ phase diameter is less than 50 nm [20]. Figure 3b is a 15,000 folds scanning electron microscopy (SEM) image. From Figure 3b, it can be seen that the primary γ′ phase is distributed on the grain boundary in a chain shape, and the secondary γ′ phase is uniformly distributed inside the grain boundary. Figure 3c is a 20,000 folds SEM image, in which the three times γ′ phase cannot be found. An energy dispersive spectrometer (EDS) analysis of the precipitates on the surface of microstructure was carried out, and the analysis results are shown in Figure 3d. The carbon content cannot be measured in the EDS analysis of the alloy. It can be judged that the material is M_23_C_6_ [16] by the presence of Cr and Mo.

An XRD (X-ray diffraction) analysis of the surface structure of FGH96 nickel-based superalloy shows that it is mainly composed of γ and γ′. The phase composition is shown in Figure 4, and the diffraction pattern shows that there are no other phases in the alloy.

### 3.2. Mechanical Property Analysis

The room temperature hardness of the FGH96 nickel-based superalloy sample was measured. To reduce the hardness measurement error and avoid accidental data, it was measured every 0.25 mm from the first point. A total of 20 points were tested, as shown in Figure 5a. It can be seen from the figure that the hardness value does not change much, indicating that the γ′ phase distribution on the alloy surface is relatively uniform. After calculating the hardness data of multiple measurements, the average room temperature hardness of the FGH96 nickel-based superalloy is 470.6 HV.

The tensile sample was tested. The relationship between material stress and strain is shown in Figure 5b. It is concluded that the tensile strength of FGH96 nickel-based superalloy is 1570 MPa and the elongation is δ 12.1%. FGH96 nickel-based superalloy relies on Ni to dissolve Cr, Co, W, Mo, and other alloy elements to strengthen the solid solution, weaken the element diffusion, form high-temperature resistant atomic groups, and reduce the stacking fault energy. At the same time, a large number of solute atoms play a pinning role in the dislocation aggregation [21] to reduce the dislocation activity. When the dislocation cuts through γ′, a higher external force is required to produce obvious second-phase strengthening. Through these two strengthening methods, the material has high tensile strength. In the plastic deformation process of FGH96 nickel-based superalloy, it mainly depends on the γ′ precipitate phase to hinder the dislocation movement. The tensile fracture morphology of FGH96 nickel-based superalloy is shown in Figure 6. In Figure 6a, the secondary crack (in the yellow ellipse) can be observed. The secondary crack grows inward perpendicular to the tensile fracture surface, which shows that the material has a certain brittleness in the tensile process. In Figure 6b, there are many equiaxed dimples, and the large number of dimples shows that the plasticity of the material is high.

### 3.3. Fatigue Performance Analysis

According to the national standard GB/T15248-2008 “metallic materials axial constant-amplitude low-cycle fatigue test method”, the fatigue test was carried out using the lifting method, and the data in Table 3 were obtained.

The data in Figure 7a were obtained by fitting the life cycle and the maximum stress from Table 3. It can be seen from the figure that most points are near the fitting curve, indicating that the test results have a good correlation. As shown in Figure 7b, the fatigue strength of 5 × 10^6^ cycles is 625.33 MPa by the linear fitting of logN and $\log S_{a}$.

Figure 8 shows the fatigue fracture morphology at 810 MPa, and the fatigue source area is shown in Figure 8a. It can be seen from the figure that the crack source is formed in the middle of the sample surface, and the crack diffuses around in a fan-shaped manner. At this time, the crack propagation rate is the slowest. Figure 8b shows the cleavage step in the fatigue source area. Figure 8c shows the stable stage of fatigue crack. There are many secondary cracks in the figure. The secondary crack shown in the yellow circle extends along the tear ridge, and the crack as a whole is Z-shaped. The secondary crack shown in the red circle passes through the tear ridge and consumes a large amount of energy, resulting in the stop of crack propagation. The fatigue stripe is a typical feature of stable crack growth. The fatigue stripe shown in Figure 8c is relatively smooth, so it is a plastic fatigue stripe, and the direction of the fatigue stripe is parallel to the direction of the secondary crack. The fatigue stripes shown in Figure 8d are brittle fatigue stripes with an uneven surface morphology, and there are cleavage steps formed by five cleavage surfaces above the stripes. Figure 8e,f show the morphology of the instantaneous fracture zone. Under a high stress cyclic load, tire indentation appears, as can be seen in Figure 8e,f, dimple morphology appears. In this stage, the crack is in the high-speed propagation phase. Under the action of fatigue cyclic load, dimples of different sizes gradually gather and connect with each other, and finally form a dimple fracture. Combined with the above characteristics, the fracture is a mix of cleavage fracture and ductile fracture.

### 3.4. Analysis of Synchrotron Radiation X-Imaging

Figure 9 shows the imaging of the SR-μCT sample with 3000 cycles under 950 MPa stress in the observation area of synchrotron radiation. The experimental data were sliced using Pitre software, and the three-dimensional image was then reconstructed using Avizo software. The volume of all defects in the sample was counted and analyzed, and the results shown in Figure 10b were obtained. There are 50 defects in the imaging area: where the volume is less than 50 μm^3^, there are 40 defects; for 50–200 μm^3^, there are six; and for more than 200 μm^3^, there are four. The sphericity of several larger defects in Figure 9 was analyzed, and the sphericity of the three defects is greater than 0.9. From the sphericity, it can be judged that the three defects are cavities. The whole defect is concentrated in the middle of the sample due to the high energy required for imaging, which reduces the total number of photons passing through the sample, resulting in the imaging results being concentrated in the middle of the sample. Figure 10a is a top view of the imaging area, and a large number of defects are distributed at the edge of the specimen. During the fatigue loading cycle, the fatigue source area is easily formed in the area where these defects are concentrated.

The fatigue test of the sample was continued, and the sample fractured when the cumulative number of cycles reached 5625. The fatigue fracture was imaged by synchrotron radiation X-ray. Figure 11b is the side view of the fracture 3D reconstruction morphology. It can be seen from Figure 11b that most of the defects in Figure 11a are surface defects. Figure 12 is a SEM diagram of the fatigue fracture of the synchrotron radiation specimen, Figure 12a is the macro morphology of the fatigue fracture, and Figure 12b is the top view of the three-dimensional morphology. It can be seen that the defects shown in the blue box of Figure 11a are in the fatigue source area of Figure 12a. As the number of fatigue cycles is less, the characteristics of the fatigue source region are not obvious, and the whole fracture morphology is similar to that of a tensile fracture. Figure 12c shows the crack growth zone, in which secondary cracks appear, and Figure 12d shows the morphology of the instantaneous fracture zone, with a large number of equiaxed dimples. Through an enlarged observation of this area, it can be seen in Figure 12e that there are complete and broken particles at the lower part of the dimple. An EDS analysis of the particles was carried out. The analysis results are shown in Figure 12f. Through its element content ratio, it can be judged that the particles are Ni_3_Al, which is an important strengthening phase of nickel-based superalloy.

To analyze this phenomenon, due to the face-centered cubic structure of Ni-based superalloy with good slip properties, the cavities grow up gradually under cyclic loading and connect with other cavities to form dimple fractures, as shown in Figure 13a. Figure 13b shows that the second-phase particles are separated from the matrix to form dimples. According to the dislocation theory, the dislocation loops are stacked around the second-phase particles under cyclic loading. When the external force is small, the dislocation loops keep equilibrium under the interaction of dislocation stacking stress and the repulsive force of the second-phase particles. When the external force gradually increases, the dislocation loops move towards the second-phase particles. When the accumulated elastic strain can overcome the interfacial bonding force between the second-phase particles or inclusions and the matrix to form a new surface, new micro voids can be formed. At the same time, the repulsive force of the second-phase particles on the dislocation ring is weakened, which makes the dislocation move to a new micro cavity under the action of external force and makes the cavity grow. With the disappearance of stacking dislocation constraints, new dislocations are generated from the dislocation sources, and the cavities expand and aggregate rapidly. Under the action of tensile stress, a large number of cavities grow up, and the cross-section of the matrix between the adjacent cavities gradually becomes smaller. When they connect with each other, fracture occurs, forming dimple fractures. The dimple formation mechanism shown in Figure 13c is similar to that in Figure 13b, except that Figure 13b shows the separation of the second-phase particles from the matrix, and Figure 13c shows the fracture of the second-phase particles. After the fracture of the second-phase particles, micro cavities are formed, which grow and gather to form dimple fractures.

Figure 14 shows two kinds of second-phase particles at the bottom of a fatigue dimple. Figure 14a shows the complete second-phase particles at the bottom of the dimple. Figure 14b shows the broken second-phase particles at the bottom of the dimple. The experimental results are consistent with the phenomenon of mechanism formation shown in Figure 13.

### 3.5. Fatigue Fracture Damage Mechanism

Figure 15a shows the cleavage planes and cleavage steps found in the fatigue test fracture. There are three cleavage planes in total. There are a large number of river patterns on one of the cleavage planes. The cleavage steps are generated by the propagation along different planes after the fatigue crack initiation, and are typical cleavage fractures.

The cleavage plane and river pattern principle is shown in Figure 15b. The fatigue crack enters the propagation stage after cracking, but it does not propagate along one plane in the propagation process, instead it will propagate along several planes at the same time. The volume of the parallel end of the fatigue test sample at the same height is 40 folds that of the SR-μCT sample. From the imaging results, it can be judged that there are a large number of small defects in the fatigue sample. When a crack encounters internal defects, grains, and the second equal interface, it may deflect and expand in different planes. The cracks in different planes extend the same distance in the same cycle, and finally form a river pattern. Therefore, a river pattern also contains steps, which are generally shallow and difficult to distinguish from the figure. When steps are formed in the process of fatigue crack propagation, some energy will be lost due to the propagation in other directions. Therefore, a large number of cracks will converge into the main cracks in the process of propagation, and the tributaries will converge into a main stream. Therefore, according to the flow direction, we can know how a crack is propagated. The tributaries of the river pattern continue to gather to form the main stream, which will eventually form cleavage steps, and the fatigue cracks will continue to expand along the cleavage plane, eventually leading to a fracture. This is the formation mechanism of the cleavage plane and cleavage steps.

In the fracture morphology, many secondary cracks are found to expand as “Z”. A model was established, as shown in Figure 16, to analyze the crack propagation mechanism. The cyclic stress of a fatigue test is represented as a sine wave, and Figure 16a shows the cyclic stress diagram of one cycle. The whole cycle was divided into five stages to analyze the Z-word expansion mechanism, as shown in Figure 16b. In the first stage, the crack is treated and closed. When the external force gradually increases, the crack begins to expand obliquely above, and then enters the second stage. In the third stage, when the external force reaches the apex, the crack tip is passivated, and the crack cannot continue to expand in the original direction due to a gradual reduction in the external force. In the fourth stage, the stress changes from tensile stress to compressive stress, the crack expands obliquely downward, and the crack tip changes into a folded ear shape under the increasing compressive stress. In the fifth stage, the compressive stress reaches maximum and the crack tip closes again until the end of this cycle. After a cycle, the crack can only advance a short distance, and this process is repeated until the sample breaks.

## 4. Conclusions

(1)The primary γ′ phase is distributed on the grain boundary in a chain shape, and the secondary γ′ phase is uniformly distributed inside the grain boundary. The average hardness of the alloy is 470.6 Hv, the tensile strength is 1570 MPa, and the elongation is 12.1%.(2)Under the condition of 5 × 10^6^ cycles, the fatigue strength of the alloy is 625.33 MPa, and the fatigue fracture is a mix of cleavage fracture and ductile fracture. In the crack growth zone, the secondary crack propagating along the tear ridge is the longest. Under the action of cyclic load and tip passivation, the crack propagates as a “Z” shape.(3)Synchrotron radiation X-ray imaging shows that a defect volume less than 50 μm^3^ accounts for 80% of the total. The dimple morphology is formed by the growth and aggregation of cavities in the material. There are three cases of cavity germination: defects of the material itself, fracture of the second-phase particles, and separation of the interface between the second-phase particles and the matrix.(4)Cleavage fracture is caused by the development of multiple cleavage planes formed by branch cracks that converge to form a main crack, the intersection of different cleavage planes to form cleavage steps, and the main crack continuing to expand forward until fracture. Cleavage fracture is the main factor in fatigue damage.(5)It is found that, compared with inclusions, voids are the main cause of fatigue fracture in FGH96 nickel-based superalloy. In future material manufacturing processes, while avoiding inclusions, the compactness of the material should be improved.

## Figures and Tables

**Figure 1 materials-14-06298-f001:**
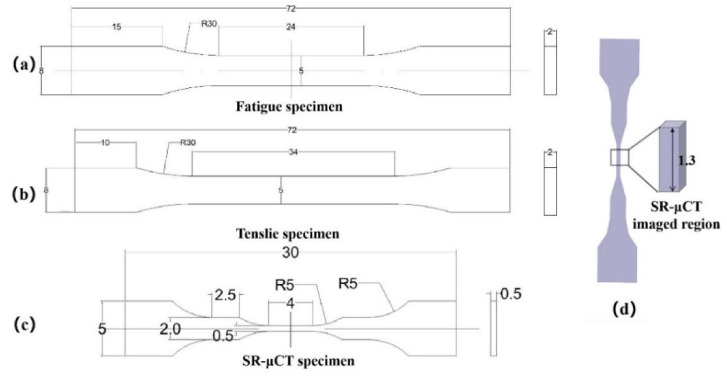
Specimen size: (**a**) fatigue specimen, (**b**) tensile specimen, (**c**) SR-μCT specimen, (**d**) synchrotron radiation imaging region (dimensions in mm).

**Figure 2 materials-14-06298-f002:**
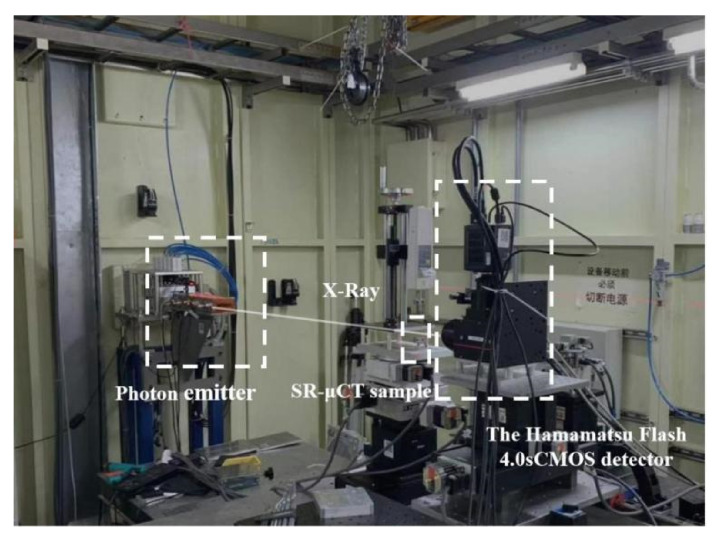
Image of the experimental device for detecting internal defects in the alloy at BL13W1.

**Figure 3 materials-14-06298-f003:**
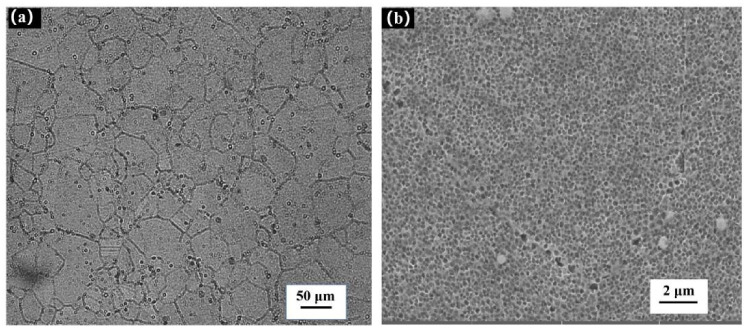
Microstructure of FGH96 alloy: (**a**) light microscope structure, (**b**) primary γ′ phase and secondary γ′ phase, (**c**) carbide M_23_C_6_, (**d**) EDS analysis results.

**Figure 4 materials-14-06298-f004:**
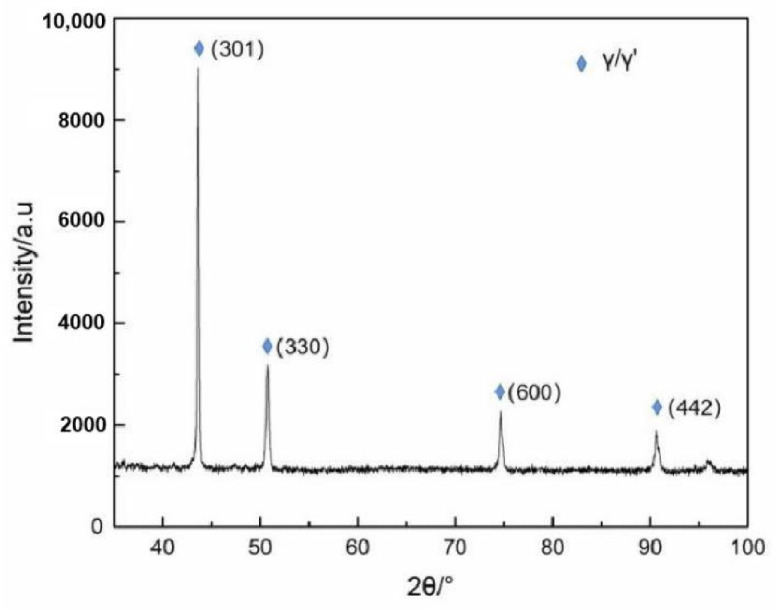
XRD diffraction pattern.

**Figure 5 materials-14-06298-f005:**
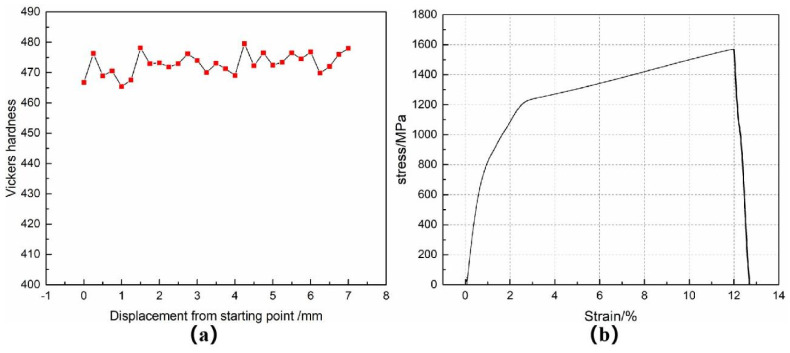
(**a**) Microhardness curve, (**b**) tensile stress–strain curve.

**Figure 6 materials-14-06298-f006:**
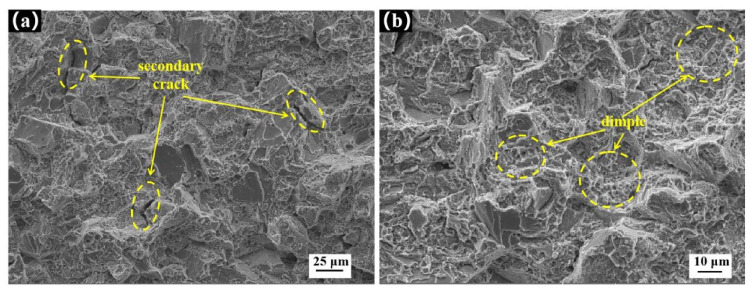
Tensile fracture morphology: (**a**) secondary cracks, (**b**) equiaxed dimples.

**Figure 7 materials-14-06298-f007:**
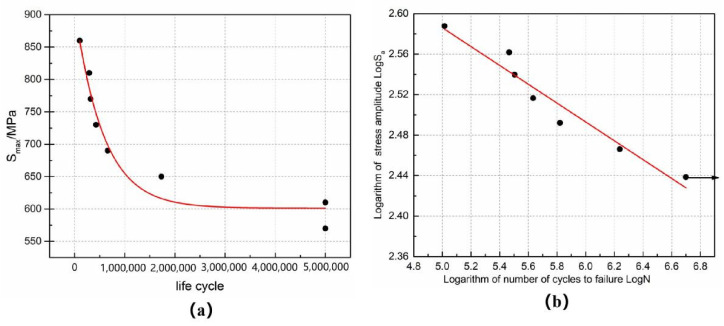
(**a**) S_max_ and life cycle curves, (**b**) LogN and LogS_a_ curves.

**Figure 8 materials-14-06298-f008:**
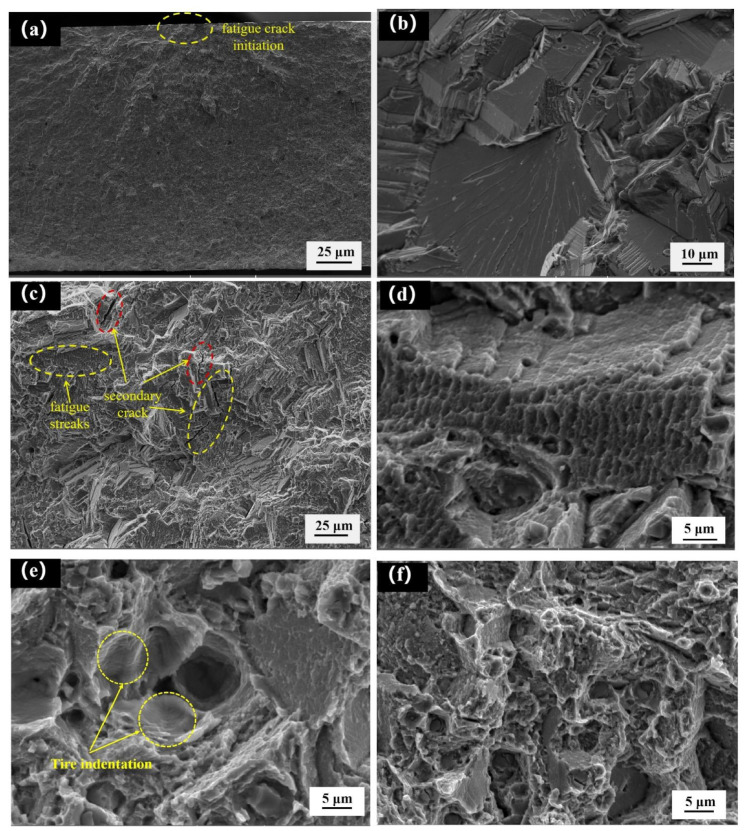
Fatigue fracture topography: (**a**) fatigue crack initiation, (**b**) cleavage step, (**c**) secondary cracks and fatigue streaks, (**d**) brittle fatigue streaks, (**e**) tire indentation, (**f**) dimples.

**Figure 9 materials-14-06298-f009:**
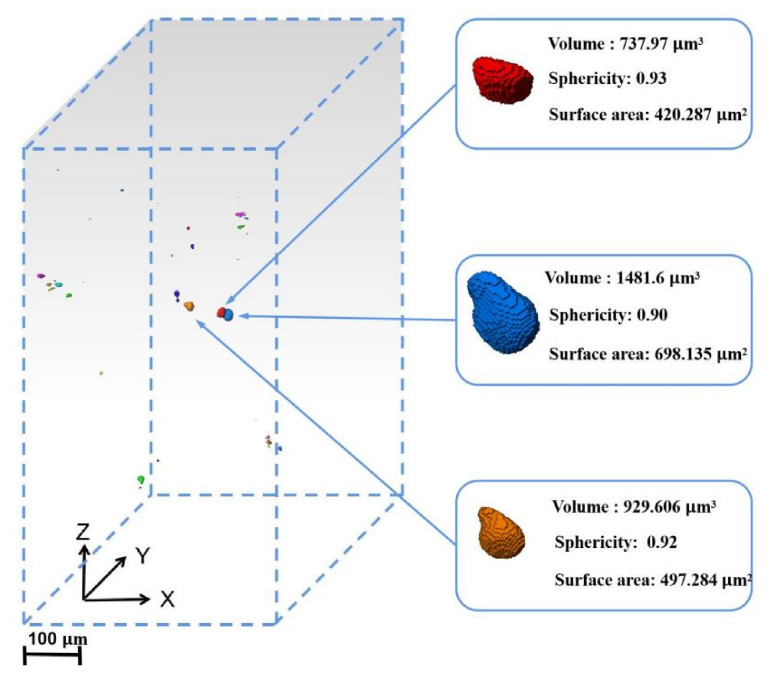
SR-μCT imaging area reconstruction showing representative defects in the micro sample.

**Figure 10 materials-14-06298-f010:**
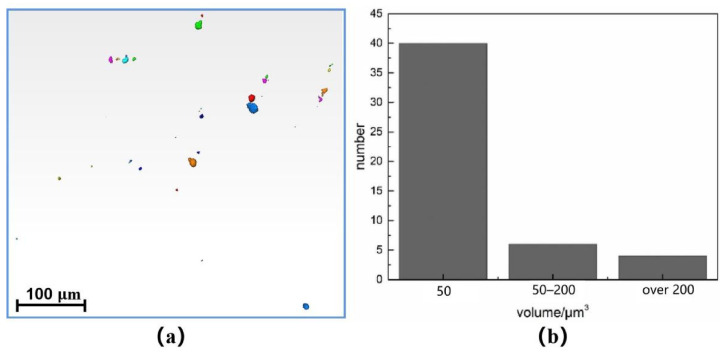
(**a**) Top view of imaging area, (**b**) statistical diagram of defect volume size.

**Figure 11 materials-14-06298-f011:**
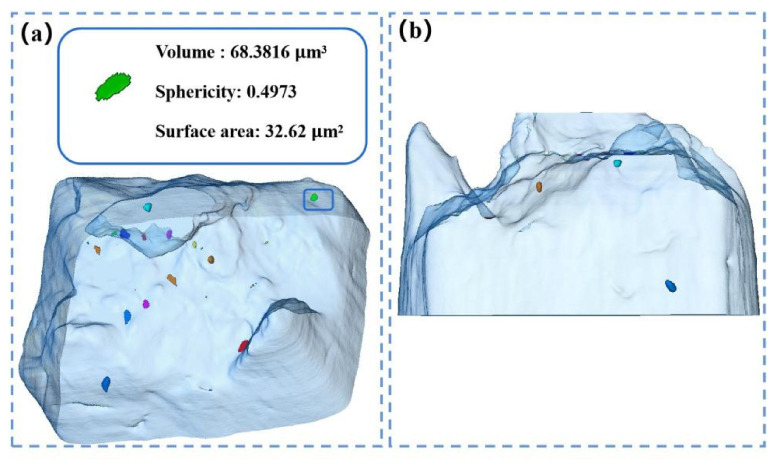
(**a**) Top view of fracture imaging, (**b**) side view of fracture imaging.

**Figure 12 materials-14-06298-f012:**
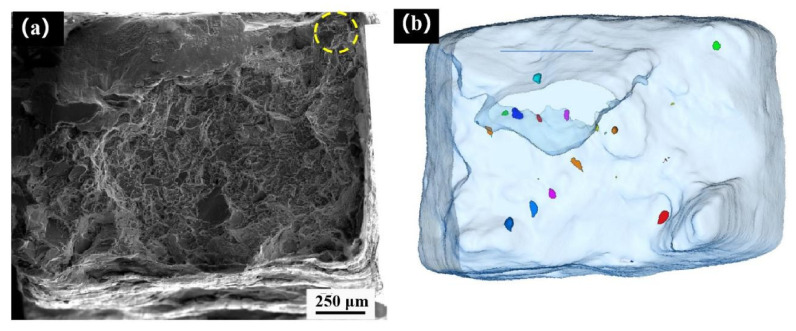
(**a**) Fatigue fracture morphology, (**b**) top view of fracture imaging, (**c**) second crack, (**d**) dimple morphology, (**e**) second-phase particles, (**f**) EDS analysis results.

**Figure 13 materials-14-06298-f013:**
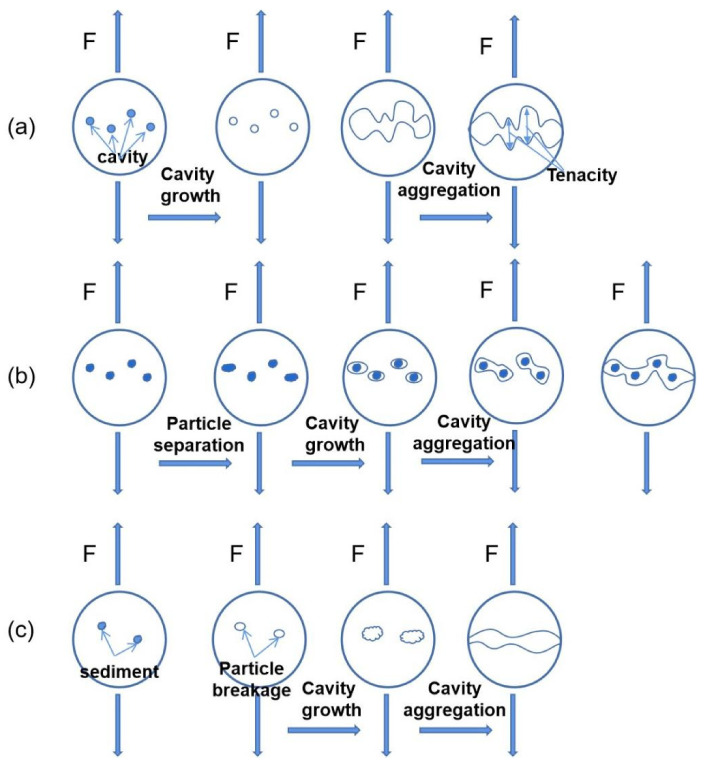
Formation mechanism of fatigue dimple: (**a**) cavity growth, (**b**) separation of second-phase particles, (**c**) fragmentation of second-phase particles.

**Figure 14 materials-14-06298-f014:**
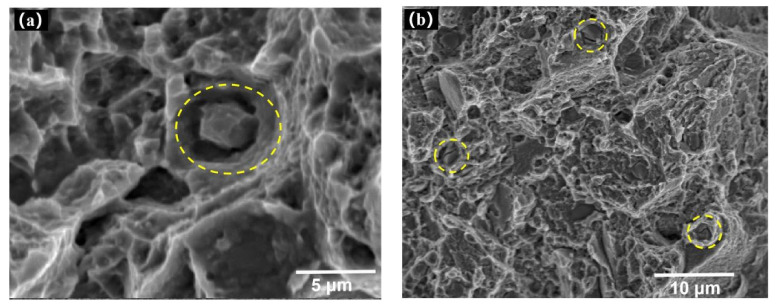
(**a**) Complete second-phase particle, (**b**) broken second-phase particle.

**Figure 15 materials-14-06298-f015:**
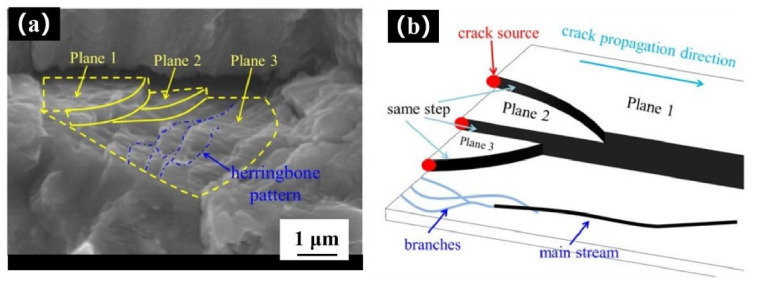
(**a**) SEM image of cleavage step, (**b**) schematic diagram of the formation mechanism of cleavage steps.

**Figure 16 materials-14-06298-f016:**
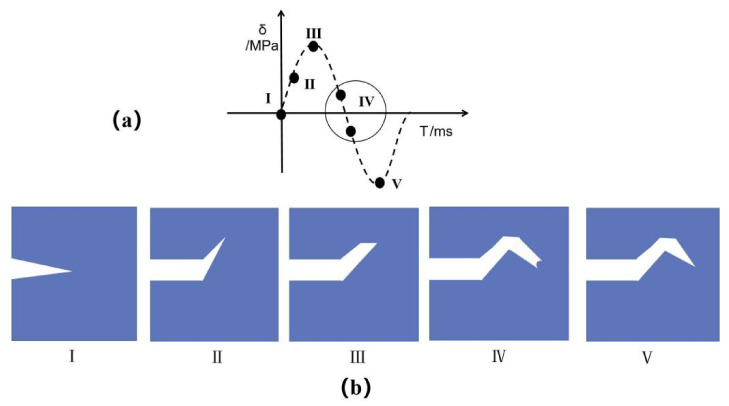
(**a**) Schematic diagram of cyclic stress, (**b**) Z-shaped expansion mechanism.

**Table 1 materials-14-06298-t001:** Chemical composition of FGH96 nickel-based superalloy (mass fraction, wt%).

Material	Mass Fraction, %											
	C	Cr	Co	W	Mo	Ce	Al	Ti	Zr	Nb	B	Ni
FGH96	0.04	15.5	13.4	3.9	3.7	0.01	2.3	3.6	0.03	0.7	0.08	Banlance

**Table 2 materials-14-06298-t002:** Test, test equipment, and test methods.

Test	Test Equipment	Test Method (Test Purpose)
Hardness testing	HV-1000 micro hardness meter (Shanghai Taiming Optical Instrument Company, Shanghai, China)	Hardness test every 0.25 mm
Microstructure observation	VHX-6000 super depth of field microscope (KEYENCE) S-3800 field emission scanning electron microscope (Japan Hitachi Co., Ltd., Tokyo, Japan)	The surface of the sample was polished and etched with corrosive agent (5 g CuCl_2_ + 100 mL HCl + 100 mL C_2_H_5_OH) for 2 min
Fatigue test	Zwick Amsler 250 HFP 5100 testing machine (ZwickRoell)	The fatigue cycle mode is sine wave, the loading frequency is 20 Hz, and the stress ratio R = 0.1
Tensile test		The tensile load is uniform until the specimen is broken
Observation of fracture morphology	S-3800 field emission scanning electron microscope	The tensile and fatigue fracture surfaces were observed and the fatigue damage mechanism was analyzed
Phase analysis	S-3800 field emission scanning electron microscope	Phase composition of alloy surface was analyzed

**Table 3 materials-14-06298-t003:** Fatigue test data.

Sample No.	Maximum Stress (S_max_/MPa)	Stress Amplitude (S_a_/MPa)	Life Cycle	Logarithm of S_a_	Logarithm of Life Cycle
1	860	387	103,477	2.5877	5.0148
2	810	364.5	291,971	2.5616	5.4653
3	770	346.5	319,143	2.5397	5.5039
4	730	328.5	429,394	2.5165	5.6328
5	690	310.5	661,253	2.4921	5.8203
6	650	292.5	1,731,470	2.4661	6.2384
7	610	274.5	5,000,000	2.4385	6.699
8	570	256.5	5,000,000	-	-

## Data Availability

The data used to support the findings of this study are included within the article.

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
