# Peer review of "Study on Microstructure and Fatigue Properties of FGH96 Nickel-Based Superalloy"

_materials, 2021, doi:10.3390/ma14216298_

Round 1
Reviewer 1 Report
Thank you very much for sending me this interesting article for review.
The following considerations should be taken into account:
- On page 6 below figure 3. Strength of ~ 625MPa is stated to occur at 5000 cycles. Shouldn't it be 5,000,000 cycles? Could the authors also plot what a single load cycle looks like? Was displacement or force control in the test?
- Was one test performed for each stress level? In the case of the 5,000,000 cycle, the results of two tests are presented. Why isn't more numer of tests needed?
- What is the Young's modulus of this material?
- How many static tests were performed to make the curve (fig. 7b)?
- Why are the samples for tension and cyclic tension slightly different? (Fig. 1a and b)?
- How were the samples prepared? How were they cut out? How did this process affect the damage distribution? Were only the SR-μCT samples prepared by wire cutting?
- What did the samples look like after breaking in static and cyclic loading tests? Did the sample crack after a limited number of cycles?
- What was the stress-strain curve for the presented cyclic tests?
- Why were the SR-μCT tests performed for higher stresses resulting in a lower number of cycles to failure?
Author Response
Response to Reviewer 1 Comments
Point 1:On page 6 below figure 3. Strength of 625MPa is stated to occur at 5000 cycles. Shouldn't it be 5,000,000 cycles? Could the authors also plot what a single load cycle looks like? Was displacement or force control in the test?
Response 1: Due to my clerical error, I wrote the fatigue strength at 625Mpa as 5000 cycles. 5000 cycles have been modified to 5 * 10 ^ 6 cycles. The single load cycle is shown in Figure 16 (b), and the whole test is controlled by force.
Point 2:Was one test performed for each stress level? In the case of the 5,000,000 cycle, the results of two tests are presented. Why isn't more numer of tests needed?
Response 2:In the process of fatigue test, each stress has been tested three times, but due to defects or fracture location, select an appropriate data from the three groups of data for fatigue life analysis. If 5,000,000 cycles are reached under both stress conditions, the material can reach the expected life at and below this stress.
Point 3:What is the Young's modulus of this material?
Response 3: The young's modulus of FGH96 nickel base alloy is 188.7Gpa
Point 4:How many static tests were performed to make the curve (fig. 7b)?
Response 4:The curve shown in Figure 7b is realized by Figure 7a. Many points in Figure 7a are on the curve, which shows that the experimental results are reasonable. The stress limit of 5000000 cycle life is obtained by linear fitting the logarithm of life and stress
The limit of stress at 5,000,000 cycles is derived by formula 1-1
1-1
N is the number of fatigue cycles, K is the fatigue strength coefficient, and C is the fatigue strength coefficient defined when the number of cycles n = 1.
1-1 remove the logarithm at both ends to obtain equation 1-2
1-2
By fitting the data, the fatigue limit formula of FGH96 nickel base superalloy can be determined as 1-3
1-3
Point 5:Why are the samples for tension and cyclic tension slightly different? (Fig. 1a and b)?
Response 5: The fatigue specimen is designed according to GB / T 3075-2008. Since the test material is part of the turbine disk, the sample can only be 72mm. In order to better calculate the tensile strain of the material, we can only reduce the distance of the clamping end and increase the distance of the parallel end.
Point 6:How were the samples prepared? How were they cut out? How did this rocess affect the damage distribution? Were only the SR-μCT samples prepared by wire cutting?
Response 6: Through processing, it is found that serious deformation will occur when the material thickness is less than 0.4mm. Therefore, the material is cut into 0.4mm flakes by wire cutting, and then cut into SR-μCT samples, and then use sandpaper to grind the sample 0.2mm thick .Due to the machining accuracy and sample size, the surface is very rough, which affects the fatigue test and reduces the fatigue life.
For fatigue samples and tensile samples, after online cutting, polish the wire cutting marks with sandpaper, and polish them with sandpaper with different roughness to avoid the influence of surface marks on the experiment
Point 7:What did the samples look like after breaking in static and cyclic loading tests? Did the sample crack after a limited number of cycles?
Response 7: Figure 1 shows the tensile fracture morphology. It can be seen from the figure that the necking phenomenon of the fracture is not obvious. It can be seen from the electron microscope that the crack starts from the surface. Under the change of material shape, the internal crack continues to expand and form a river pattern. Obvious shear lip appears around the fracture.
Figure 2 shows the macro morphology of the fatigue fracture sample. No other cracks are found on the sample surface, which shows that the sample cracks propagate from the inside. The specimen also has no obvious deformation.
This kind of sample was not found in the whole fatigue test that cracks appeared in the sample after a limited number of cycles. There are two reasons why this experimental result is not produced. First, the sensitivity of the experimental quipment is not enough, and the crack can not be detected in time. The experiment is still carried out until the fracture occurs. Second, due to the high cyclic stress, once the crack surface expands, the fracture will occur rapidly.
.
Figure 1 Tensile fracture morphology
Figure 2 Macro morphology of fatigue specimen
Point 8:What was the stress-strain curve for the presented cyclic tests?
Response 8: Please provide your response for Point 8. (in red)
Perhaps you refer to figure 7a stress and life curve, or Figure 16a cyclic stress diagram. In this paper, there is no stress-strain curve of cyclic test, only the stress-strain curve of tensile test. The stress-strain curve means that after a certain number of cycles, the change of stress-strain tends to be stable, and the hysteresis loop is close to the curve of closed loop. It is usually used to analyze the effect of cyclic hardening or softening. Figure 7a shows the cycle life and cycle stress curve, through which the fatigue strength limit under a certain cycle life can be inferred. Figure 16a shows the transformation of cyclic stress in one cycle, so as to analyze crack growth
Point 9:Why were the SR-μCT tests performed for higher stresses resulting in a lower number of cycles to failure?
Response 9: SR-μCT samples are processed by wire cutting, and then polished by sandpaper to obtain the thickness required for the test. Due to the size of the sample, the side of the sample cannot be polished well, so there are some wire cutting marks. Under high stress cyclic loading, these wire cutting traces become the place of stress concentration. Fatigue failure starts from the part with high stress or strain, forms damage and gradually accumulates, and finally leads to failure.
Thank you for your valuable suggestions on my article.

Reviewer 2 Report
This paper presents the results of a well-performed experimental study with some applied value. The information received is quite new and useful, so the paper will be of interest to some readers.
Two recommendations for authors to improve the text of the article.
1. Edit the text more thoroughly. It contains many absence of necessary spaces between characters and words. In addition, the names of the figures in some cases are written with a lowercase letter, which is incorrect.
2. It is better to remove the words "In this paper" at the beginning of the abstract.
3. The conclusions should show a broader perspective of using the results of the study. That is, I propose to supplement the information received with an indication of how to use it to improve product performance.
Author Response
Response to Reviewer 2 Comments
Point 1:Edit the text more thoroughly. It contains many absence of necessary spaces between characters and words. In addition, the names of the figures in some cases are written with a lowercase letter, which is incorrect
Response 1:
Thank you for your careful and valuable comments on the details of my article.
The text has been thoroughly edited and some missing spaces between characters and words have been added. Meanwhile, Modify the abbreviation of figures' names.
Point 2:It is better to remove the words "In this paper" at the beginning of the abstract.
.
Response 2:
According to your opinion.The word "in this article" at the abstract has been deleted.
Point 3:The conclusions should show a broader perspective of using the results of the study. That is, I propose to supplement the information received with an indication of how to use it to improve product performance
Response 3:
Thank you very much for your suggestions. According to the experimental results, I added a section to the conclusion.
It is found that compared with inclusions, voids are the main cause of fatigue fracture of FGH96 nickel base alloy. In the later material manufacturing process, while avoiding inclusions, the material compactness should be improved.
Thank you for your valuable suggestions on my article.

Round 2
Reviewer 1 Report
Thank You for possibility of the reading this interesting article. In my opinion Authors should add the units to axis in Fig16a. Reader should know what are the displacement and time duration of the single cycle in all experimental cases. It will help If somebody will want to repeat these experiments or to simulate it. The article is accepted form my side but Authors really should add these informations.